# Effect of Particle Size on Mechanical Property of Bio-Treated Sand Foundation

**Defeng Yang, Guobin Xu * and Yu Duan**

State Key Laboratory of Hydraulic Engineering Simulation and Safety, Tianjin University, 135 Yaguan Road, Jinnan District, Tianjin 300350, China; yangdf@tju.edu.cn (D.Y.); duanyu09@tju.edu.cn (Y.D.)

* Correspondence: xuguob@tju.edu.cn; Tel.: +86-22-2740-1127

**Abstract:** In the field of geotechnical engineering, microbially induced calcium precipitation technology is feasible and sustainable alternative to improve the engineering characteristics of sand foundation under different geological conditions for a long time. However, it is unclear how the effects of different sand particle sizes on the engineering characteristics of bio-treated sand column. The method of intermittent injection in batches was used to develop a series of bio-treated sand columns. The results showed that the mechanical properties of the bio-treated column improved by increasing the particle size. Low concentration of bacterial suspension and cementation reagent leads to the increase of calcium carbonate and unconfined compressive strength. Additionally, the total injection times increased, thus risking time cost. Furthermore, the increase of sand particle size was beneficial to the uniformity of the spatial distribution of calcium carbonate in the bio-treated column. The coefficient of variation was reduced by up to 52.0%. Scanning electron microscopy results confirmed that the size and uniformity of calcite crystals on the surface of sand particles were related to the concentration of cementation solution.

**Keywords:** bio-treatment; sand foundation; sand particle size; calcium carbonate contents; unconfined compressive strength

## 1. Introduction

The traditional foundation treatment of mechanical preloading, cement, or chemical grouting has exposed their limitations in the current infrastructure construction [1–3]. Mechanical preloading can only strengthen surface soil, and the treatment methods of cement and chemical grouting are at a disadvantage for its short flow distance, long consumption time, and serious environmental pollution. Consequently, more and more attention has been paid to environmentally friendly and sustainable soil improvement techniques [4].

The advantages of cementing loose sand with microbially induced calcite precipitation (MICP) technology, such as lower permeability coefficient, more environmental protection, controllable reaction process, and long transmission distance, have been studied in depth [5–9]. MICP is a widely existing biological mineralization process in nature [10]. Microbial mineralization involves complex biochemical reactions. Hydrolysis of urea is catalyzed by urea-producing bacteria attached to the surface of sand particles to produce carbonate ions ($CO_3^{2-}$) and ammonium ions ($NH_4^+$), and then the $CO_3^{2-}$ combines with the externally injected calcium ions ($Ca^{2+}$) to form a cementitious calcium carbonate precipitation on the surface of the loose sand particles. Loose sand particles are solidified to form a building material with a certain mechanical strength [10], making it possible for soft sand foundation to be modified and solidified in situ by bio-treatment [11–13]. Moreover, it is a promisingly sustainable construction material [14].

In order to make the best of MICP technology in engineering practice, the most effective treatment method and process optimization are critical to guide the use of MICP technology under different geological conditions. Soon et al. [9] explored the viability of the MICP technique for improving the engineering properties of a typical tropical residual soil, and the effectiveness of MICP was evaluated by optimizing the treatment conditions. Rebata-landa [15] reported that the ideal size range was 50–400 μm for bio-treatment. Li and Tian [16] solidified the desert aeolian sand, primarily composed of sand grains with diameters of 100–250 μm, the feasibility and effectiveness of use of this process was then evaluated based on the density, permeability, calcium carbonate content, unconfined compressive strength, and microstructure of the treated sand. Malcolm [12] investigated that the biomineralization tests were manipulated on potentially liquefiable saturated riparian soil (The particle size range of 91–210 μm) in the laboratory and field. The biomineralized soils showed properties that indicate calcite precipitation increased their resistance to seismic-induced liquefaction. Peng [17] conducted a series of test using ureolytic bacteria to reinforce the organic clay (The particle size range of 12–99 μm), with cementation reagent with different concentrations flowing to the soil under pressures. After treatment, the unconfined compression strength can increase by up to 370%, and the permeability coefficient can be reduced by about one order of magnitude. Mortensen et al. [18] tested the bio-treated columns on a wide range of grain sizes comprised of sand, silty sand, and silt and concluded that the MICP treatment was equally robust for these soils. Additionally, the effect of particle size on the mechanical behavior of other building materials was also reported. Haddock et al. [19] evaluated the impact of aggregate gradation on asphalt mixture performance. They used two mixture types having nominal maximum aggregate sizes (NMAS) of 9.5 mm and 19.0 mm to evaluate the sensitivity of asphalt mixture performance to gradation changes. The effect of particle-size distribution on the one-dimensional compressive behaviour of granular soil materials was investigated using the discrete-element method (DEM) by Minh et al. [20]. Pouranian et al. [21] reported that an asphalt mixture's aggregate skeleton, related to voids in the mineral aggregate (VMA), is another important factor that affects critical asphalt mixture properties such as durability, workability, permeability, rutting, and cracking resistance.

The biomineralization technology can effectively reinforce sand foundation with different geological conditions. Based on this, this paper attempted to solidify loose sandy soil with different particle sizes, and considers the influence of the concentration of bacteria and cementation reagent on its mineralization effect. The reinforcement effect of the process was evaluated by the number of operation cycles, unconfined compressive strength and calcium carbonate content. The morphology of calcium carbonate in the bio-treated column was analyzed by means of XRD and SEM to elucidate enhancement mechanisms. This study advances the bio-treated process towards practical engineering by addressing geological condition hurdles faced with during the upscaling process.

## 2. Materials and Methods

### 2.1. Materials

#### 2.1.1. Microorganism and Growth Conditions

The urease positive microorganism used was *Sporosarcina pasteurii* (strain ATCC 11859), purchased from the American Type Culture Collection. A liquid medium, which contained yeast extract of 20 g/L, ammonium sulfate of 10 g/L, and Tris reagent of 17.5 g/L, was used to cultivate bacteria. Individual ingredients of the recipe were separately dissolved in distilled water bottles and autoclaved separately by autoclaving for 20 min at temperature of 121 °C and then combined post-sterilization. All operations were carried out under aseptic condition after sterilization of liquid medium. Suspended bacterial cultures were cultured in a shaker at a temperature of 30 °C and a rotational speed of 220 rpm until the stable period (Figure 1C) of the bacterial growth characteristic curve. The microorganism growth characteristic curve is shown in Figure 1. The $OD_{600}$ value (the optical density of the sample at a wavelength of 600 nm) and urease activity of the bacterial suspension were measured following the

methods utilized by Harkes et al. [22]. The $OD_{600}$ value was regarded at the bacterial concentration in this study. Other desired $OD_{600}$ value was obtained by dilution with sterile sodium chloride solution (9 g/L NaCl). The relative urease activity was determined from the change in conductivity of the media due to the generation of ammonia and carbonate ions [23].

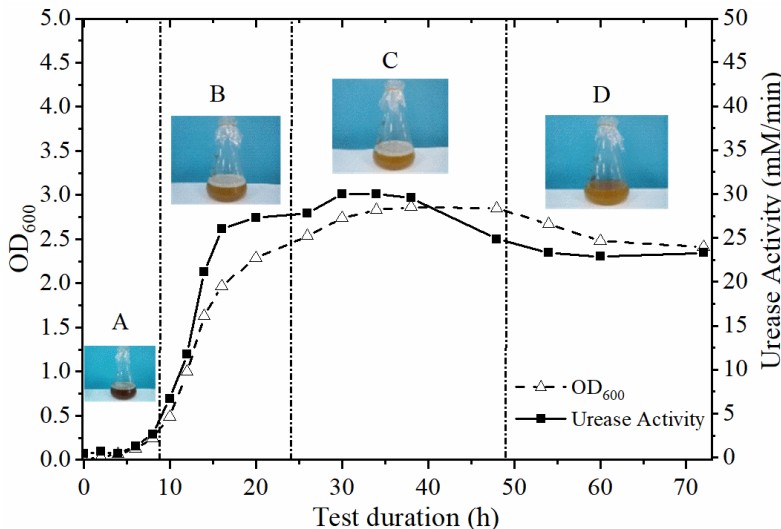

**Figure 1.** The microorganism growth characteristic curve: (**A**) adjustment period; (**B**) logarithmic period; (**C**) stable period; (**D**) decay period.

### 2.1.2. Cementation Reagent

Large amounts of $CO_3^{2-}$ and $Ca^{2+}$ are required during the crystallization of calcium carbonate. Urea can be hydrolyzed to $CO_3^{2-}$ and $NH_4^+$ under the catalysis of microbial metabolites urease, and its reaction process has good controllability. Water-soluble calcium chloride crystals can provide a large number of $Ca^{2+}$. Therefore, urea and calcium chloride were selected as the inducer and calcium source of mineralization reaction respectively [24,25]. The cementation reagent for the microbially induced calcite precipitation treatment consisted of equimolar concentrations of urea and calcium chloride [9].

### 2.1.3. Sand Specimen

The quartz sand was used to prepare columnar specimen for the bio-treatment from natural river sand in Tianjin, China. Table 1 tabulates the values of the physical indices of the three kinds of sand specimen with different particle size obtained from the standard soil properties tests. Their particle size distribution is presented in Figure 2.

**Table 1.** The values of the physical indices of the quartz sand.

| Particle Size Range $d$ (μm) | 75–150 | 150–300 | 300–600 |
|:---|:---:|:---:|:---:|
| Apparent density $\rho'$ (g/cm$^3$) | 2.644 | 2.647 | 2.650 |
| Loose bulk density $\rho_0'$ (g/cm$^3$) | 1.241 | 1.329 | 1.398 |
| Compact bulk density $\rho$ (g/cm$^3$) | 1.451 | 1.494 | 1.532 |
| Carbonate content (%) | 0.08 | 0.26 | 0.31 |

### 2.2. Laboratory Setup

The schematic diagram of the experimental setup, intermittent injection in batches, is drawn in Figure 3. The apparatus consisted of power system, cementitious system, and drainage system. The purpose of the power system was to pump the bacterial suspension and the cementation reagent into the cementitious system, which was composed of a beaker, a peristaltic hose, a peristaltic pump, and a cementing system adapter. The liquid flow was directed from top to bottom by peristaltic pump.

The cementitious system was the reaction chamber where the mineralization occurred. The sand columns had a height to diameter aspect ratio of 2:1. The interior of the plexiglass tube (wall thickness 6 mm) was used as a sand room with a diameter of 50 mm and a height of 100 mm. The buffer mesh was placed at the top of the column, and the anti-leak sand mesh was pressed at the bottom of the room to minimize possible loss of sand particles during liquid injection. The tube was positioned vertically with rubber plug upper surface, and a hose clamp was connected to the base. The drainage system was used to collect effluent.

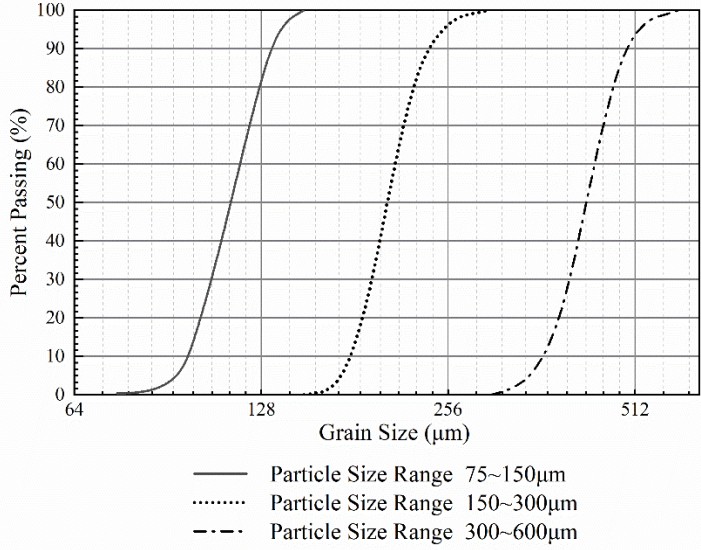

**Figure 2.** Particle size distribution of three kinds of sand specimen.

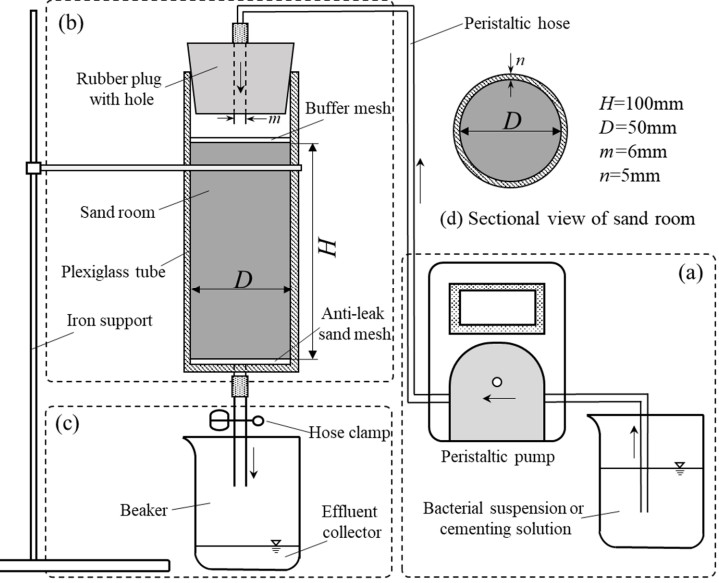

**Figure 3.** Schematic diagram of the microbially induced calcite precipitation (MICP) experimental setup: (**a**) power system; (**b**) cementitious system; (**c**) drainage system; (**d**) sectional view of sand room.

## 2.3. Experimental Procedure

### 2.3.1. Mixed Horizontal Orthogonal Experimental Design

The bio-cemented process was restricted by various factors [26]. To prepare the best possible reaction condition, the experimental environmental factors were set up on the basis of existing

scientific research. All treatments were performed at room temperature (25–30 °C) [27] and controlled pH 6.5–8.5. The purpose of this experiment was to carry out the bio-treated sand test under the coupling condition of particle size, concentration of bacterial suspension and cementation reagent. Bacterial suspension and cementation reagent were alternately injected multiple times in the bio-cemented process. The unconsolidated granular medium was reinforced with gelatinous calcium carbonate ($CaCO_3$) from mineralization reaction to form a material of certain mechanical strength [28].

Orthogonal experimental design is a scientific method to arrange and analyze multi-factor tests by orthogonal tables [29]. The level of orthogonal experimental factors is shown in Table 2. It was assumed that each parameter is non-interactive during this reaction. The $L_{12}(3 \times 2^4)$ mixed horizontal orthogonal test table was used to design the orthogonal test scheme. The test design scheme groups are reviewed in Table 3. The number of matches at different levels was equal, and the method can reflect objective laws with a high calculation efficiency.

**Table 2.** The horizontal values of the factors in the orthogonal test.

| Factor Level | $d$ (mm) | $C_b$ $OD_{600}$ | $C_c$ (mol·L$^{-1}$) |
|:---:|:---:|:---:|:---:|
| 1 | 0.075–0.15 | 0.5 | 0.5 |
| 2 | 0.15–0.3 | 2 | 1.5 |
| 3 | 0.3–0.6 | - | - |

**Table 3.** $L_{12}(3 \times 2^4)$ mixed horizontal orthogonal test table.

| Scheme | Column Number | | | | | Test Index | | |
|:---:|:---:|:---:|:---:|:---:|:---:|:---:|:---:|:---:|
| | $d$ (mm) | $C_b$ $OD_{600}$ | $C_c$ (mol·L$^{-1}$) | Empty Column | Empty Column | $\lambda$ | $q_u$ (kPa) | $\bar{\delta}$(%) |
| 1 | 0.075–0.15 | 0.5 | 0.5 | 1 | 1 | 8 | 352 | 13.13 |
| 2 | 0.075–0.15 | 0.5 | 0.5 | 2 | 2 | 7 | 584 | 12.20 |
| 3 | 0.075–0.15 | 2.0 | 1.5 | 1 | 2 | 3 | 127 | 10.07 |
| 4 | 0.075–0.15 | 2.0 | 1.5 | 2 | 1 | 3 | 458 | 11.40 |
| 5 | 0.15–0.3 | 0.5 | 1.5 | 1 | 1 | 5 | 155 | 12.40 |
| 6 | 0.15–0.3 | 0.5 | 1.5 | 2 | 2 | 4 | 313 | 10.48 |
| 7 | 0.15–0.3 | 2.0 | 0.5 | 1 | 1 | 7 | 165 | 12.03 |
| 8 | 0.15–0.3 | 2.0 | 0.5 | 2 | 2 | 8 | 1368 | 14.64 |
| 9 | 0.3–0.6 | 0.5 | 1.5 | 1 | 2 | 7 | 216 | 12.37 |
| 10 | 0.3–0.6 | 0.5 | 0.5 | 2 | 1 | 10 | 1369 | 16.30 |
| 11 | 0.3–0.6 | 2.0 | 0.5 | 1 | 2 | 7 | 219 | 9.77 |
| 12 | 0.3–0.6 | 2.0 | 1.5 | 2 | 1 | 4 | 903 | 14.28 |

### 2.3.2. Sand Biochemical Treatment Cycles

The loose sand was loaded into the sand room of the experimental device in Figure 3 in three times. The filled soil in the sand room was shaken to form a 100-mm-high sand specimen until compact bulk density itself.

First, the specimens were percolated in reverse direction at flow rate of 1 mL/min by distilled water (DW, 200% of PV of the granular specimens) until ponding on top of the specimen. This step was to wet the sand columns and release the gas from itself. Second, bacterial suspension (BS, 150% of PV of the granular specimens) was percolated at flow rate of 5 mL/min [30]. Percolation of cementation reagent (CR, 150% of PV of the granular specimens) at flow rate of 10 mL/min was performed after the previous step was standing for 3 h. Again, percolation of cementation reagent was done in the same way after a 5 h retention time. The fluid was repeatedly injected into the sand chamber in this way until the bottom of cementitious system no longer oozed out, and the biochemical treatment cycles was complete. The injection of once BS and twice CR was a bio-treated batch. The total batch number of biological treatments was recorded as $\lambda$.

### 2.3.3. Sampling and Geotechnical Laboratory Measurement

After completion of the bio-treatments, the specimen mold was dismantled carefully and the sand column was extruded from the mold. Then all the specimens were dried in a constant temperature oven for 48 h at 30 °C, and trimmed to a height of 100 mm and diameter of 50 mm for the unconfined compression strength (UCS) test in accordance with ASTM 2166 [31]. The displacement-loaded axial rate was carried out at a constant rate of 0.5 mm/min, as shown in Figure 4. The UCS values for bio-treated soil specimens were denoted as $q_u$. Noted that a thin layer of sand was laid between the column and the equipment support to make a flat contact surface during this test.

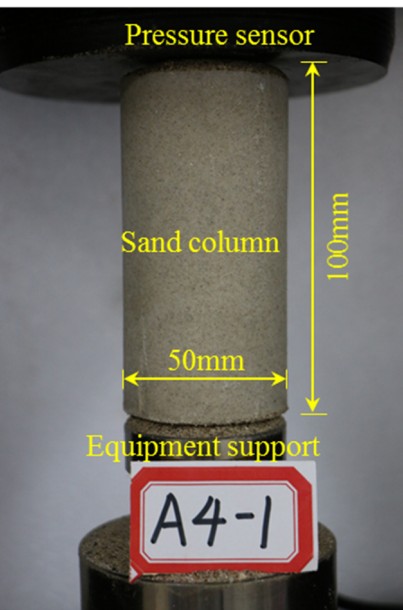

**Figure 4.** Setup of unconfined compression test.

### 2.3.4. Quantification of Calcium Carbonate

After completion of the geotechnical laboratory measurement, the calcium carbonate content was measure using an excess hydrochloric acid washing (1.5 M HCl) method [32]. For this purpose, the bio-treated subsamples were cut into 5 equal portions along the horizontal direction, as shown in Figure 5. Each part of the subsamples was crushed sufficiently and dried in an oven at a temperature of 105 °C, and then their masses were measured before and after the acid wash (1.5 M excessive hydrochloric acid solution for 24 h). Noted that the pH of the residual solution was verified using a test paper. The calcium carbonate content was the reduced mass of the solid particle washed with sufficient distilled water and dried at the same way because it would be dissolved in the acid solution. The percentage of calcium carbonate of subsamples was obtained by the following equation:

$$\delta_i = \frac{m_0 - m_1}{m_0} \times 100\% \tag{1}$$

where $\bar{\delta}$ is the quantification of calcium carbonate ($i$ = 1, 2, 3, 4, 5, %), $m_0$ is the initial mass of the subsamples (10 g), and $m_0 - m_1$ is the reduced mass of the solid particle (g).

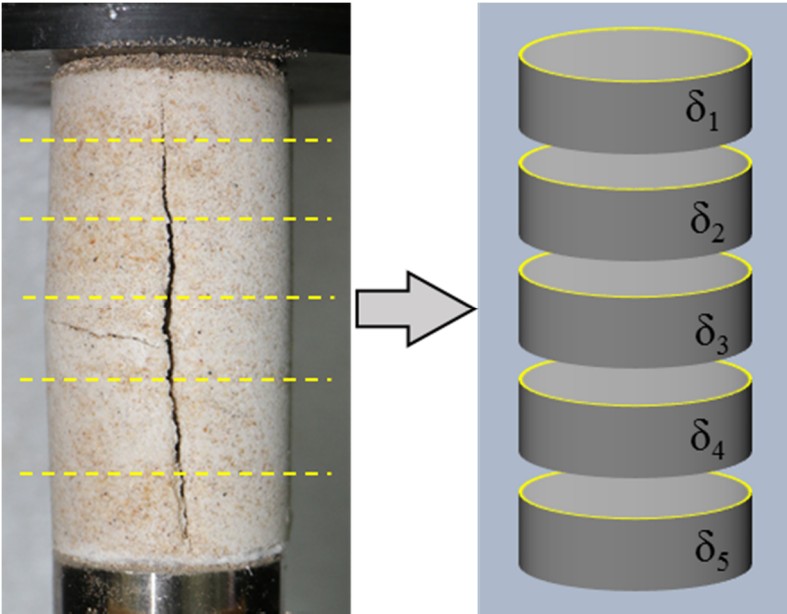

**Figure 5.** The layering diagram of the bio-treated subsamples along the horizontal direction.

Several indicators were defined to determine the distribution of calcium carbonate in sand column. The average value of the percentage of calcium carbonate $\bar{\delta}$ was calculated as

$$\bar{\delta} = \sum_{i=1}^{n} \delta_i / n \tag{2}$$

where $n$ is the number of layers. The coefficient of variation $C \cdot V$ can be expressed as follows:

$$C \cdot V = \frac{\sqrt{\sum_{i=1}^{n} (\delta_i - \bar{\delta})^2 / n}}{\bar{\delta}} \tag{3}$$

### 2.3.5. Mineralogical and Microscopic Imaging

A D/MAX-2500 X-ray diffraction (XRD) analyzer was used to identify the new elements and minerals formed in microbially treated samples. Over-dried untreated sand particles (particle size 150–300 μm) and bio-treated sand column (Scheme 7: 150–300 μm, $OD_{600}$ = 2, $C_c$ = 0.5 M) were used for XRD analysis with Cu-K$\alpha$ radiation at operating voltage of 40 kV and current of 30 mA. Then, the analysis results were contrasted with standard diffraction patterns from the International Centre for Diffraction Data. Modification in the micro-structure and formation of new chemical composition were selected and observed by scanning electron microscopy (SEM, model number: S-4800). Over-dried powered samples in the scheme of Section 2.3.1 were sputter-coated with platinum before testing. The SEM test parameters were as follows: EHT = 3.00 kV, WD = 6 mm.

## 3. Results and Discussion

### 3.1. Direct Analysis of Orthogonal Experiment

Each test condition was completed in strict accordance with the prescribed scheme. The results of each metric test and calculation are shown in Table 4. The results of the above mixed level orthogonal test were analyzed by multi-index range analysis to obtain the influence degree of each factor on the bio-cemented column and the optimal level combination. Three parameters were introduced into the orthogonal experimental intuitive analysis method, respectively, namely, $K_{jm}$, $k_{jm}$, and $R_j$ ($j$ = 1, 2, 3,

4 and 5; $m$ = 1, 2 or 3). The sum of the $j$th-factor and $m$th-level test result ($\lambda$, $\bar{\delta}$, $q_u$) was recorded as $K_{jm}$. $k_{jm}$ represented the average of the $j$th-factor and $m$th-level test result, as shown in Equation (4).

$$k_{jm} = K_{jm}/s \qquad (4)$$

where $s$ is the number of occurrences of the $m$ level in the $j$th-column. $R_j$ represents the $j$th-column range and reflects the degree of influence of an experimental factor on the index results, as shown in Equation (5).

$$R_j = \max\{k_{j1}, k_{j2}, k_{j3}\} - \min\{k_{j1}, k_{j2}, k_{j3}\} \qquad (5)$$

**Table 4.** The values of $K_{jm}$, $k_{jm}$, and $R_j$ at each factor level.

| Index | | $d$ (mm) | $C_b$ OD$_{600}$ | $C_c$ (mol·L$^{-1}$) |
|---|---|---|---|---|
| | $k_{j1}$ | 5.25 | 6.83 | 7.83 |
| | $k_{j2}$ | 6.00 | 5.33 | 4.33 |
| $\lambda$/times | $k_{j3}$ | 7.00 | - | - |
| | $R_j$ | 1.75 | 1.50 | 3.5 |
| | Primary and secondary order | | $C_c > d > C_b$ | |
| | Preferred embodiment | | $C_{c1}d_3C_{b1}$ | |
| | $k_{j1}$ | 380 | 540 | 676 |
| | $k_{j2}$ | 500 | 498 | 362 |
| $q_u$/kPa | $k_{j3}$ | 675 | - | - |
| | $R_j$ | 295 | 42 | 314 |
| | Primary and secondary order | | $C_c > d > C_b$ | |
| | Preferred embodiment | | $C_{c1}d_3C_{b1}$ | |
| | $k_{j1}$ | 11.63 | 12.78 | 13.73 |
| | $k_{j2}$ | 12.39 | 12.02 | 11.82 |
| $\bar{\delta}$/% | $k_{j3}$ | 13.18 | - | - |
| | $R_j$ | 1.55 | 0.76 | 1.91 |
| | Primary and secondary order | | $C_c > d > C_b$ | |
| | Preferred embodiment | | $C_{c1}d_3C_{b1}$ | |

The greater the $R_j$ value is, the greater the influence of the level of experimental factors on the outcome indicators will be. The intuitive analysis results of multi-factor coupling on the test indicators are shown in Table 4, which described the primary and secondary influencing factors and the preferred scheme. As shown in Table 4, the primary and secondary order of the influencing factors in $\lambda$, $q_u$ and $\bar{\delta}$ was consistent, all of which were $C_c > d > C_b$. The optimal scheme for the test indexes was $C_{c1}d_3C_{b1}$ (1, 2, and 3 are the level numbers of each factor).

The variation pattern of $k_{jm}$ value of each test factor in Table 4 with level is shown in Figure 6. From Figure 6a, within the range of sand particle size, the $k_{jm}$ values of total infusion times $\lambda$, unconfined compressive strength $q_u$ and calcium carbonate content $\bar{\delta}$ increased gradually with the increase of $d$, indicating that the increase of particle size was conducive to the improvement of solidified sand strength. With the continuous mineralization reaction, the pores between particles were filled with mineralized products, and the permeability coefficient of the sample tended to decrease gradually until the injected liquid was difficult to permeate inside the sample, which was regarded as the end of the mineralization test. As it is clear from Table 1, the compact bulk density of sand particles with particle sizes (75–150 µm, 150–300 µm, and 300–600 µm) are 1.451 g/cm$^3$, 1.494 g/cm$^3$, and 1.532 g/cm$^3$, respectively. The initial void volume of sand column increased with the increase of particle size level, and the amount of calcium carbonate required for filling also aggrandized accordingly, thus increasing the required times of liquid injection. Therefore, the injection cycle times $\lambda$ of sand with different particle sizes is related to the compact bulk density $\rho$ of sand particles. The higher the initial compact bulk density, the more times of liquid injection are required, correspondingly the longer the mineralization

reaction time, and the fuller the pores between sand particles are filled with calcium carbonate, which eventually leads to the rising unconfined compressive strength of bio-treated column samples.

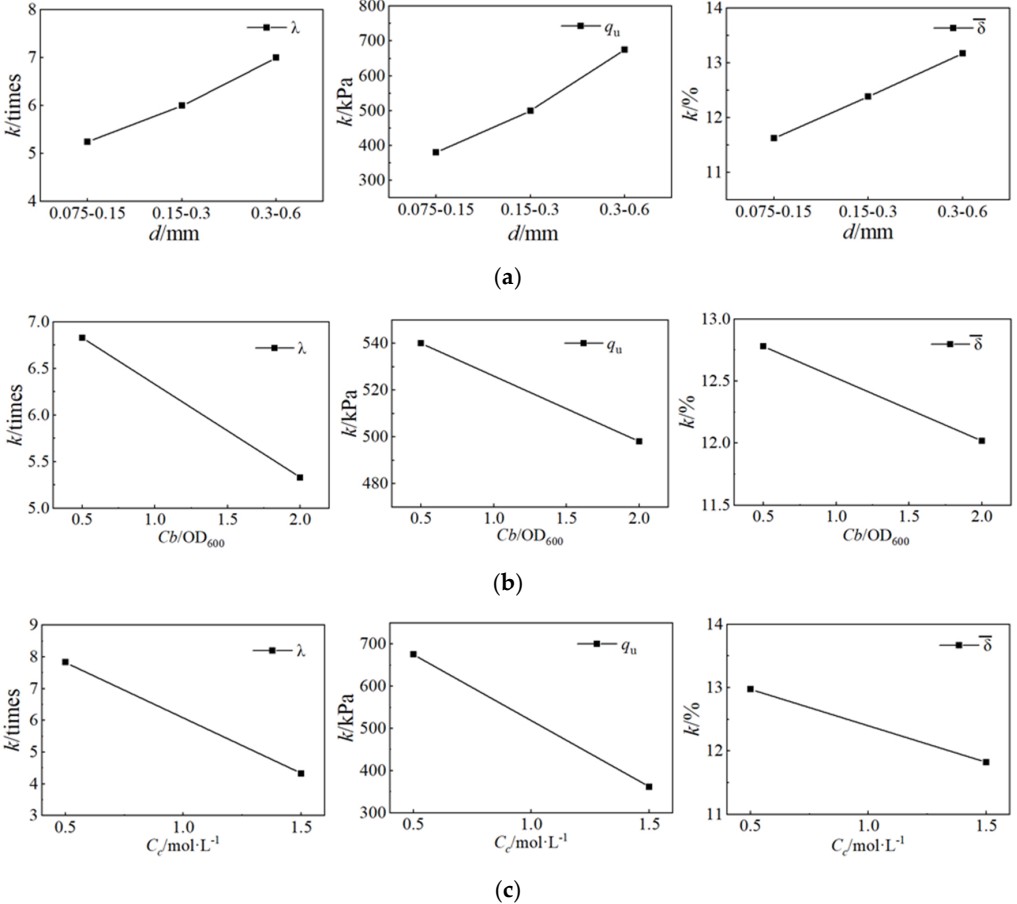

**Figure 6.** The variation pattern of $k_{jm}$ value of each test factor with level: (**a**) effects of sand particle size on test index; (**b**) effects of concentration of bacterial suspension on test index; (**c**) effects of concentration of cementation reagent on test index.

The $k_{jm}$ values of $\lambda$, $q_u$ and $\bar{\delta}$ decreased with the increase of concentration of cementation reagent $C_c$ within the test range, as shown in Figure 6b. In the process of grouting and curing, the higher concentration of cement solution can provide sufficient mineralization raw materials for the formation of inchoate calcium carbonate, so the gap of sand particles was filled quickly by the generated calcium carbonate from the mineralization reaction. Due to the limitation of the vertical injection operation, the pores in the upper part of the sand column were filled rapidly. The number of total fluid injection was reduced on account of pore plugging in the upper part of sand column [33]. Not only that, a high concentration of cementation reagent may obstruct the formation of calcium carbonate. A large amount of $Ca^{2+}$ will be adsorbed on the surface of bacterial cells, which will affect the hydrolysis of urea by bacterial urease [23]. This also means that the adoption of solutions with lower concentration of cementation reagent may require more fluid injection times and thus more time cost in the actual operation.

The $k_{jm}$ values of $\lambda$, $q_u$ and $\bar{\delta}$ also decreased with the increase of concentration of bacterial suspension $C_c$ within the test range, as shown in Figure 6c. This indicates that the high concentration of bacterial suspension is not conducive to the increase of calcium carbonate content in the overall column and the improvement of unconfined compressive strength of the sand column.

### 3.2. The Spatial Distribution of Calcium Carbonate Precipitation in the Columns

The uniformity of the material is a nonnegligible factor in the application of bio-treated sand foundation as an improvement technology [32], as it can affect the targeted strength. The distribution of calcium carbonate precipitation along the height of bio-treated granulometric columns for the current study is shown in Figure 7. It was obvious that the calcium carbonate precipitated more in the upper portion of sand column than any other part of these column. This is because the areas closest to bio-treatment were more easily exposed to mineralized cementation reagent than the other region of the cylinder [22]. Furthermore, this increased precipitation deposition on the upper surface may relate to the fact that the newly injected microbial cells were adsorbed by the upper sand particles, which led to a reduction in concentration of bacteria along the injection path [34].

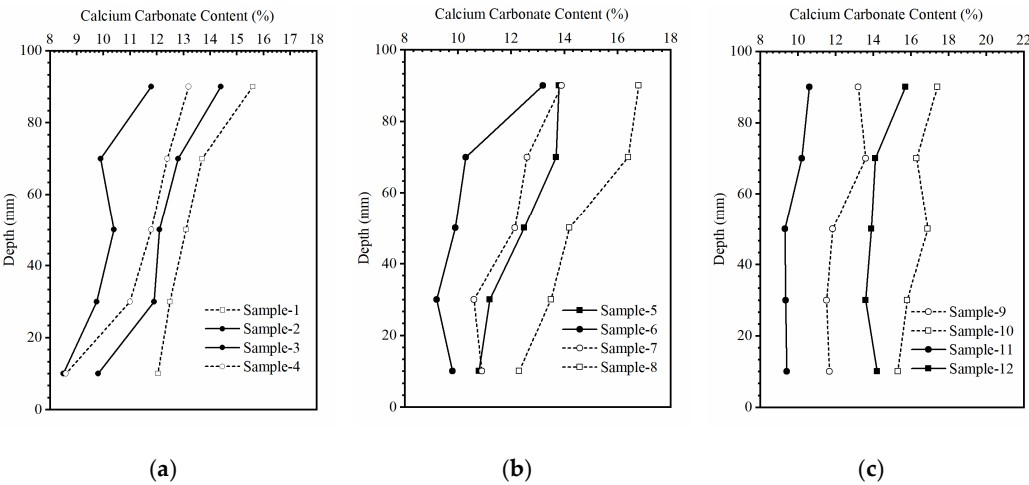

**Figure 7.** Quantification calcium carbonate contents along the height of solidified sand specimens: (**a**) 75–150 μm; (**b**) 150–300 μm; (**c**) 300–600 μm.

The calcium carbonate crystals in the whole sand column were unevenly distributed along the vertical direction. On the whole, the content of calcium carbonate in the sand column decreased from top to bottom, and the plugged phenomenon at the upper part of the column did exist during the bio-treated process (mentioned in Section 3.1). Quantitative analysis of each bio-treated column showed that the maximum range R of calcium carbonate content in the column along different heights was 4.6, 4.4, and 2.1%, respectively, accounting for 40.4, 30.1, and 12.9% of the average value in three groups of tests with different particle size ranges of Figure 7a–c. By the sand column of particle size of 300–600 μm, the maximum range R of calcium carbonate content was decreased by approximately 54.3% compared with the particle size of 75–150 μm.

The coefficient of variation (C·V) of calcium carbonate contents was found to be, respectively, 10.3, 13.6, 11.9, and 15.4% for sample 1, 2, 3, and 4 in the Figure 7a. The C·V was found to be, respectively, 11.2, 13.3, 10.2, and 13.1% for sample 5, 6, 7, and 8 in the Figure 7b. The average C·V decreased by 6.7% compared with the particle size of 75–150 μm, where the C·V was found to be, respectively, 7.7, 5.1, 6.1, and 5.7% for sample 9, 10, 11, and 12 in the Figure 7c, and the average C·V decreased by 52.0% compared with the particle size of 75–150 μm.

In the present study, the calcium carbonate distribution in consolidated column is more uniform for groups with larger particle size, which is expected to meet the requirements of soil foundation treatment technology. This is because it has the largest pore volume which allows easier flow for reagent solution and bacterial cells to occur. Previous studies had demonstrated that uneven deposition of calcium carbonate resulted in heterogeneity in strength distribution of engineering properties of improved sand foundation [35]. Therefore, the breakthroughs should be made in technology and injection methods. It is suggested that vertical grouting technology should be improved to horizontal grouting technology.

### 3.3. The Relationship between the UCS and CaCO₃ Content

Under unconfined compression strength conditions, the failure modes of sand columns were divided into tensile failure, shear columns, and tensile-shear composite failure [36]. As shown in Figure 8, the test results of the UCS of MICP-treated sand samples presented that almost all the failure modes were tensile–shear composite failure, which was mainly manifested as the damage had legible tensile failure surface and shear-slip surface. The main crack was dominated by the tensile failure surface.

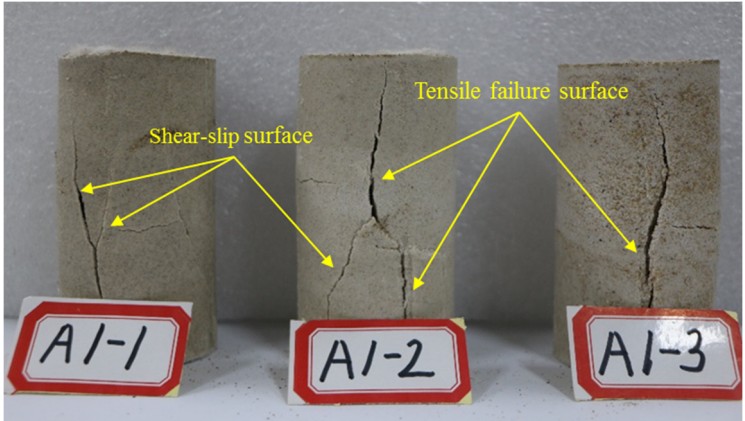

**Figure 8.** Failure mode of unconfined compressive strength test of bio-treated sand column.

The relationship between unconfined compressive strength $q_u$ and calcium carbonate content in the column is presented in Figure 9. The unconfined compressive strength $q_u$ improved with the increase of average calcium carbonate content in the sand column, which verified the research of Qabany [37] and Duo [16] that calcium carbonate content was positively correlated with the mechanical strength of the bio-treated sand column. The calcium carbonate content in the solidified sand column can be used to estimate the UCS value $q_u$ of the sample to some extent.

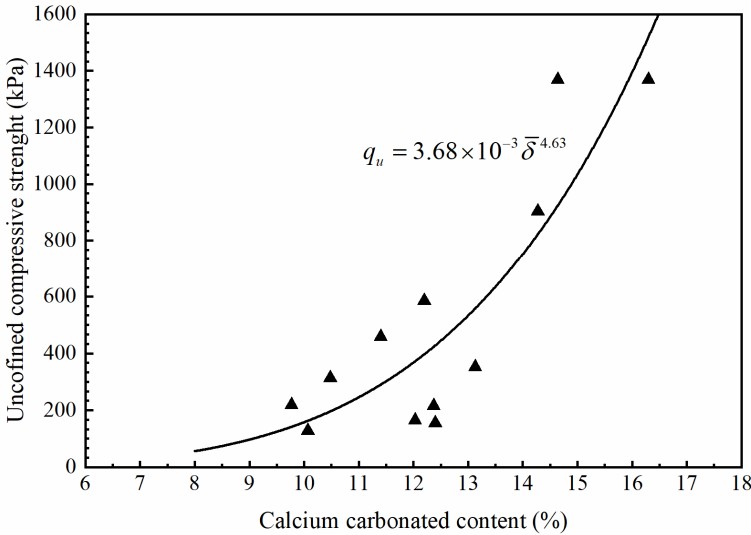

**Figure 9.** Relationship between the unconfined compressive strengths and average calcium carbonate content.

Figure 9 plots the relationship between the unconfined compressive strengths and average calcium carbonate contents of the specimens. The theory of Harkes [38] and Cheng [39] suggested that the cementation between sand particles cannot offer strong support at low calcium carbonate levels.

The UCS of the column improved rapidly with the increase of calcium carbonate content when the cementation can begin to support sand particles. Therefore, the relationship between unconfined compressive strength $q_u$ and calcium carbonate content in the column can be fitted by an exponential function (6).

$$q_u = 3.68 \times 10^{-3} \bar{\delta}^{-4.63} \tag{6}$$

### 3.4. Mineralogical and Crystal Morphology Analysis

The XRD analysis and SEM observation provided trustworthy evidences on the precipitation of calcium carbonate from MICP mineralization. Figure 10a,b illustrate XRD spectrum peaks intensity of diffraction angle for untreated and bio-treated sand specimen, respectively. The main components of the tested sand column were quartz ($SiO_2$) and calcite ($CaCO_3$), respectively. The calcium bicarbonate $Ca(HCO_3)_2$ and ammonium chloride ($NH_4Cl$) are the other products that could potentially form, besides calcium carbonate. Quartz was the main mineral component in the original sand, and calcite was a new cementitious chemical composition obtained after bio-treatment with MICP technology, which was also the fundamental reason why the bio-treated column gained a certain mechanical strength [7,9]. This was done to confirm that the crystals observed in the images were actually $CaCO_3$ precipitated on the silica sand and to detect whether there were any other elements in the column samples after the precipitation process.

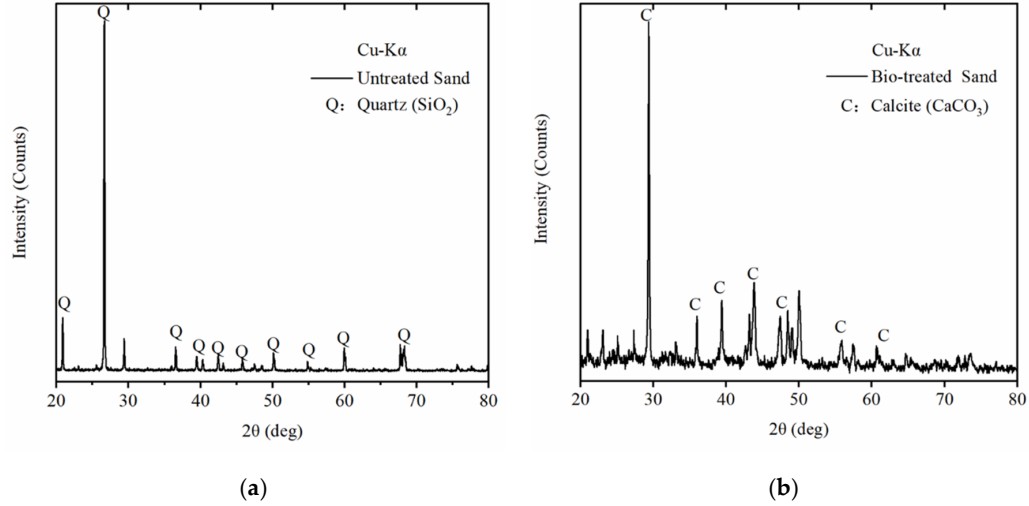

(**a**)             (**b**)

**Figure 10.** (**a**) XRD spectrum peaks for untreated sand specimen; (**b**) XRD spectrum peaks for bio-treated sand specimen.

Based on the performance of the above macroscopic mechanical tests, lower concentration of cementation reagent and more injection cycles resulted in samples with more pores filled, higher calcium carbonate content and greater mechanical strength. This indicated that the concentration of cementation reagent had a potential effect on the microstructure of sand column. Differences in the distribution of calcium carbonate crystals (sand particle size: 300–600 μm) were observed obviously on SEM image, as shown in Figure 11a–f. Each sample was photographed several times in order to exclude the randomness of experimental observation. As shown in Figure 11a,b, the original sand particles had a smooth surface. Both bio-treated specimen at cementation reagent of 0.5 M and 1.5 M showed that the abundant calcite ($CaCO_3$) crystals were filled at the contact point and covered the surfaces of the sand particles.

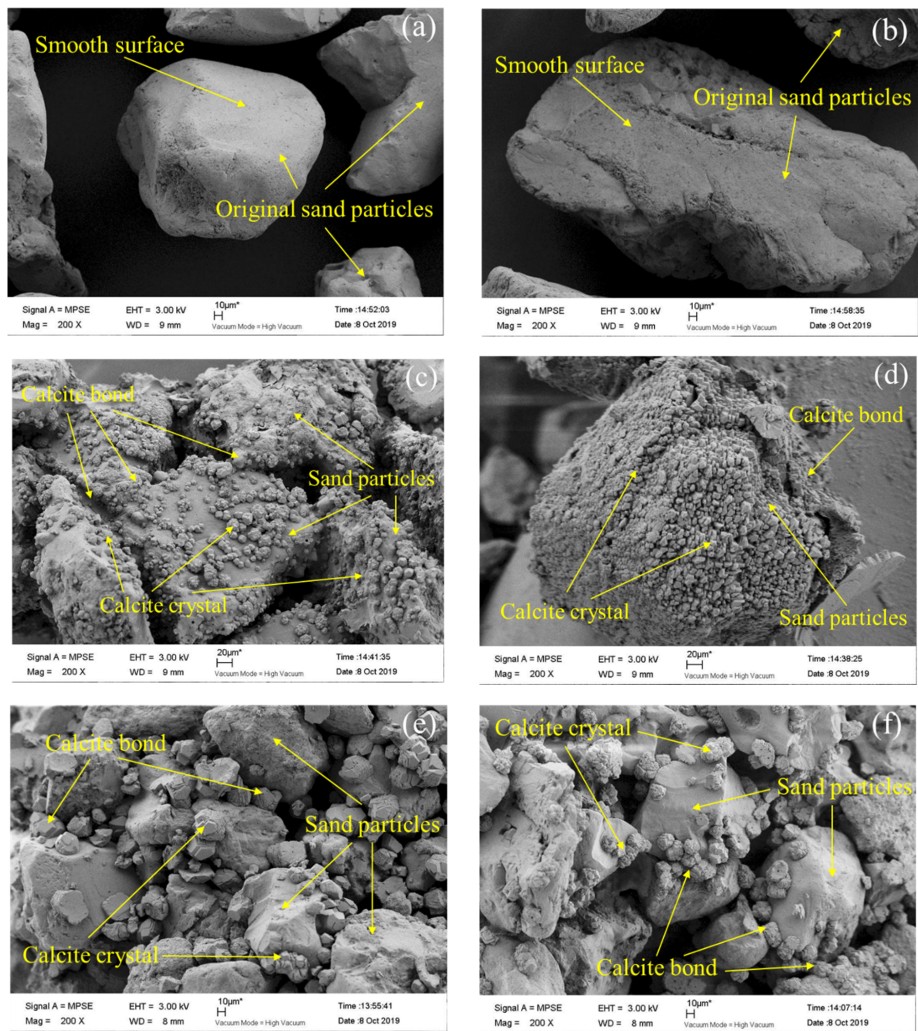

**Figure 11.** SEM micrographs of untreated and bio-treated column specimen magnified 200×: (**a**,**b**) original sand particles specimen; (**c**,**d**) bio-treated specimen with lower concentration (0.5 M Urea+CaCl$_2$); (**e**,**f**) bio-treated specimen with higher concentration (1.5 M Urea+CaCl$_2$).

Comparatively, the calcite crystal size was smaller, denser, and uniformly distributed on the surface of sand particles for the bio-treated specimen of lower concentration (0.5 M Urea+CaCl$_2$), which was visualized qualitatively in Figure 11c,d. In the bio-treated specimen with the higher concentration of cementation reagent (1.5 M Urea+CaCl$_2$) in Figure 11e,f, the calcite crystals grew in heaps and gathered on the original crystal, resulting that the crystals were larger in size and more dispersed in distribution. The phenomenon was consistent with the conclusion reported by Qabany and Soga [37]. Crystallization from solution can be considered a two-step process. The first step is the phase separation or "birth" of new crystals. The second step is the growth of these crystals to larger sizes. These two processes are known as nucleation and crystal growth, respectively [35]. Von Weimarn rules summarized the experimental observation of the average crystal size from supersaturated solution and revealed the dependence of the average crystal size on the relative concentration of supersaturated solution [40,41]. Somain et al. [42] reported that the higher concentration of Ca$^{2+}$ and CO$_3^{2-}$ in the reaction solution, the larger average particle size of the precipitate (CaCO$_3$). The consumption of CO$_3^{2-}$ mainly acted on the nucleation of calcite, rather than promoting the growth of calcite crystals at low concentration of Ca$^{2+}$ and CO$_3^{2-}$.

These findings provide insights into the understanding of the inorganic substances in the reactants mentioned above in terms of microscopic observation. To better understand the strength

enhancement mechanisms, it is necessary to study that the fine and coarse aggregates are mixed in various percentages. Additionally, the subject of correlation mechanism between the microstructure and mechanical properties of bio-treated sand bodies is likely to have a more detailed discussion in the future.

## 4. Conclusions

A series of mixed levels orthogonal experiments were conducted to examine the effect of particle size range, bacterial suspension concentration, and cementation fluid concentration on efficacy of bio-treated sand columns. The following conclusions can be drawn from this study:

- In the present research, the increase of sand particle size is beneficial to improve the effect of bio-treated sand, which may be related to the compact bulk density of sand particles. At low concentration bacterial suspension and low concentration cementation reagent, more calcium carbonate is formed, which increases the unconfined compressive strength.
- As the sand particle size increased, the uniformity of calcium carbonate precipitation was better along its height of the bio-treated column. The coefficient of variation was reduced by up to 52.0%.
- Smaller and denser calcium carbonate crystals are precipitation in the pore matrix when using low concentration cementation reagent for bio-treated loose sand particles.

**Author Contributions:** Conceptualization, D.Y. and G.X.; methodology, G.X. and Y.D.; validation, D.Y. and G.X.; formal analysis, Y.D.; investigation, D.Y. and Y.D.; resources, G.X. and Y.D.; data curation, D.Y.; writing—original draft preparation, D.Y.; writing—review and editing, D.Y. and G.X.; visualization, D.Y.; project administration, G.X.; funding acquisition, G.X. All authors have read and agreed to the published version of the manuscript.

**Funding:** This research received no external funding.

**Acknowledgments:** All workers from the State Key Laboratory of Hydraulic Engineering Simulation and Safety of Tianjin University are acknowledged. The authors are also grateful for the assistance of the anonymous reviewers.

**Conflicts of Interest:** The authors declare no conflict of interest.

## Abbreviations

The abbreviations are used in this manuscript:

| | |
|---|---|
| MICP | Microbially induced calcite precipitation |
| $OD_{600}$ | The optical density at a wavelength of 600 nm |
| $d$ | Particle size range (μm) |
| $\rho$ | Compact bulk density (g/cm$^3$) |
| $\rho'$ | Apparent density (g/cm$^3$) |
| $\rho_0'$ | Loose bulk density (g/cm$^3$) |
| H | The height of the sand room (mm) |
| D | The diameter of the sand room (mm) |
| $m$ | The diameter of the tube for injecting liquid (mm) |
| $n$ | The wall thickness of the plexiglass tube (mm) |
| $C_b$ | Concentration of bacterial (mol·L$^{-1}$, M) |
| $C_c$ | Concentration of cementation reagent |
| BS | Bacterial suspension |
| DW | Distilled water |
| CR | Cementation reagent |
| PV | Pore volume (cm$^3$) |
| $\lambda$ | The total batch number of biological treatments (times) |
| $q_u$ | The unconfined compression strength of bio-treated soil specimens (kPa) |
| $\bar{\delta}$ | The average calcium carbonate content (%) |
| R | Rang |
| $C{\cdot}V$ | The coefficient of variation |
| XRD | X-ray diffraction |
| SEM | Scanning electron microscopy |
| Q | Quartz |
| C | Calcite |

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
