# Peer review of "Effect of Particle Size on Mechanical Property of Bio-Treated Sand Foundation"

_applsci, doi:10.3390/app10228294_

Round 1

Reviewer 1 Report

The paper deals with microbially induced calcite precipitation in sand samples with the aim of ground reinforcement. The results presented demonstrates that particle sizes are influential in this regard, and a higher value of particle size is a plus in obtaining better results in stabilization.

The article is well structured and the results are well presented, although in the conclusion section, future directions might as well be added.

Some small observations are as follows:

- I suggest not using abbreviations in the abstract section;

- line 41: bio-treatment instead of bio-treated;

- line 75: first word I think should be bacteria and not bacterial

- space between results and their measuring units should be used all over the paper;

- figure 5 - the presented schematic is horizontal, not vertical direction;

- figure 6 could be removed since it brings no novelty or value to the article;

- lines 372-373: please rephrase.

Author Response

Dear Reviewers:

Thank you for the reviewer’s comments concerning our manuscript (ID: applsci-989126). Those comment are all valuable and very helpful for revising and improving our papers, as well as the vital guiding significance to our researches. We have studied comments carefully and have made correction which we hope meet with approval. Revised portion are marked in red in the paper.

Reviewer 2 Report

This work is trying to evaluate the effects of different sand particle sizes on the engineering characteristics of bio-treated sands. The following comments are suggested to improve this manuscript: 1- Line 10: In the abstract please use the full name of MICP because it is the first time it is used. 2- For the introduction, please use more previous related works as a background. The authors also can mention the effect of particle size on the mechanical behavior of other materials such as asphalt mixture and cementitious materials. for example, the following work “Determination of voids in the mineral aggregate and aggregate skeleton characteristics of asphalt mixtures using a linear-mixture packing model” 3- The writing needs to be improved as some part of this work is really hard to understand what exactly the authors are trying to say. 4- Please improve the quality of Fig 7. 5- The authors showed that the relationship between the unconfined compressive strengths and average calcium carbonate content is exponential. Is there any behind reason or theory why this trend was observed? 6- Please improve the conclusion section. It looks like a summary.

Author Response

(The authors gave the same response as above.)

Reviewer 3 Report

The study refers to the consolidation of sand specimens employing a bio-treatment procedure that yields calcium carbonate. The concept is application-oriented that makes the work very interesting.

  1. The language of the text has to be revised. There are several mistkaes, especially in parts of section 3.4.
  2. Please explain the abbreviations when they appear first (e.g. in Abstract).
  3. The authors used XRD analysis to characterize the consolidation products. Please mention which are the other products that could potentially form, besides calcium carbonate. Please add this information in lines 330-331. Please explain, then, why you observed only calcium carbonate.
  4. Lines 362-367 are not necessary, since a summary of the findings fits better to the Conclusions section.
  5. Please make more clear the suggestions for future work (lines 368-373).
  6. Please include in the Conclusions section only the main findings of the study.   

Author Response

(The authors gave the same response as above.)

Round 2

Reviewer 2 Report

This manuscript has been significantly improved after revising. So it can be published as it is

This manuscript is a resubmission of an earlier submission. The following is a list of the peer review reports and author responses from that submission.